# Field Performances of Mediterranean Oaks in Replicate Common Gardens for Future Reforestation under Climate Change in Central and Southern Europe: First Results from a Four-Year Study

**Filippos Bantis** [1,*][iD], **Julia Graap** [1], **Elena Früchtenicht** [1][iD], **Filippo Bussotti** [2][iD], **Kalliopi Radoglou** [3] and **Wolfgang Brüggemann** [1,4]

1   Department of Ecology, Evolution and Diversity, Goethe University Frankfurt, Max-von-Laue-Str.13, D-60438 Frankfurt, Germany; julia.graap@hotmail.de (J.G.); e.fruechtenicht@gmx.de (E.F.); w.brueggemann@bio.uni-frankfurt.de (W.B.)
2   Department of Agri-Food Production and Environmental Science, University of Florence, Piazzale delle Cascine 28, I-50144 Florence, Italy; filippo.bussotti@unifi.it
3   Department of Forestry and Management of the Environment and Natural Resources, Democritus University of Thrace, Pantazidou 193, GR-68200 Nea Orestiada, Greece; kradoglo@fmenr.duth.gr
4   Senckenberg Biodiversity and Climate Research Center, Senckenberganlage 25, D-60325 Frankfurt, Germany
*   Correspondence: fbanths@gmail.com

**Abstract:** Climate change imposes severe stress on European forests, with forest degradation already visible in several parts of Europe. Thus adaptation of forestry applications in Mediterranean areas and central Europe is necessary. Proactive forestry management may include the planting of Mediterranean oak species in oak-bearing Central European regions. Five replicate common gardens of Greek and Italian provenances of *Quercus ilex*, *Q. pubescens* and *Q. frainetto* seedlings (210 each per plantation) were established in Central Italy, NE Greece (two) and Southern Germany (two, including *Q. robur*) to assess their performance under different climate conditions. Climate and soil data of the plantation sites are given and seedling establishment was monitored for survival and morphological parameters. After 3 years (2019) survival rates were satisfactory in the German and Italian sites, whereas the Greek sites exerted extremely harsh conditions for the seedlings, including extreme frost and drought events. In Germany, seedlings suffered extreme heat and drought periods in 2018 and 2019 but responded well. Provenances were ranked for each country for their performance after plantation. In Greece and Italy, *Q. pubescens* was the best performing species. In Germany, *Q. pubescens* and *Q. robur* performed best. We suggest that Greek or Italian provenances of *Q. pubescens* may be effectively used for future forestation purposes in Central Europe. For the establishment of *Quercus* plantations in Northern Greece, irrigation appears to be a crucial factor in seedling establishment.

**Keywords:** *Quercus*; morphology evaluation; survival rate; extreme frost; heat and drought

## 1. Introduction

Climate change scenarios bring forward particularly challenging effects on European forests leading to the necessary development of new strategies for the maintenance of functional forests. Extreme temperatures and severe drought events are predicted to increase in frequency, like it has previously been recorded in 2018 and 2019 across Europe [1,2]. Environmental changes can be so abrupt that natural replacement of forest tree species with better adapted species may not occur at sufficient speed to prevent severe forest degradation. Southern regions may lose large areas of their forest cover altogether, except for their higher mountain ranges. The total net financial loss of the value of European forest land from these shifts is estimated to be up to hundreds of billions Euro in the period to 2100 [3], while ecological implications are also substantial.

Proactive forestry management may include the planting of Mediterranean oak species in oak (*Quercus robur* L., *Q. petraea* Matt.) bearing Central European regions [4,5]. This approach aims at the establishment of (at least) seed donor and (at best) timber plantations of future climate-tolerant species within existing oak stands to ensure the continuous presence of closely related oak species, thus maintaining oak-dominated ecosystem functions including oak-associated biodiversity. Such a concept is currently realized e.g., in the South Hesse Oak Project (SHOP [5]). However, since Central Europe and low/medium elevation forests in the Mediterranean will face–in principle–the same problems in the forthcoming century, this approach may be extended to forest management strategies in mountain regions in Mediterranean countries as well.

One way of testing the response of species or populations under certain environmental conditions is by establishing common gardens in several locations of ecological interest [6]. Common gardens are considered essential in determining the potential of selected species and/or provenances [7] to be implemented for 'assisted migration' due to climate change [8–10].

For reforestation purposes, different strategies can be applied. One option is to start the process by active seed dispersal, usually leading to healthy plants with well-developed root systems, especially in tap rooting species like most oaks [11]. However, buried seeds are often prone to herbivory in the field (e.g., Pausas et al. [12]). Therefore, foresters, especially in Germany, often rely on pre-growth of seedlings in tree nurseries and transplanting established containered seedlings into the field sites. This strategy minimizes the need for seed material, but implies additional working steps and costs. To further minimize potential losses, various methods of fast and easy selection of vigorous seedlings have been suggested (for review see e.g., Mattson [13]). For example, in the case of the Mediterranean oak species *Quercus ilex* and *Q. coccifera*, Tsakaldimi et al. [14] tested the use of morphological parameters like height, leaf area or stem diameter as easily measurable parameters for the prediction of seedling survival in the first year after outplanting. Since Mattsson [13] also suggested the use of chlorophyll content and fluorescence as potential, easily measurable predicitive parameters, we attempted to verify their usefulness for the prediction of first-year survival in the field under different climate conditions.

The objectives of the present study were to evaluate the survival and growth responses of seedlings from three oak species established on experimental plantations exposed to different environmental conditions including climate, water availability and soil properties. Specifically, *Quercus ilex* L., *Quercus frainetto* Ten., and *Quercus pubescens* Willd. from Greek and Italian provenances were grown in replicate common gardens in Southern Germany, Central Italy, and Northern Greece. The German plantations also include a local oak species, *Quercus robur* L. for comparative purposes.

The specific questions addressed were:

(1) Do the three Mediterranean species differ in survival and growth performance during the initial growth phase under different macroclimatic conditions?
(2) Do Greek and Italian provenances of the Mediterranean species (consistently) show different behavior in different environments?
(3) Do the local German oaks (*Q. robur*) outperform their Mediterranean relatives in the German plantations?
(4) Is it possible to predict survival and performance in the field from easily accessible morphological and physiological seedling parameters?

## 2. Materials and Methods

### 2.1. Plant Material

In autumn 2015, seeds of three common Mediterranean oak species, *Q. ilex*, *Q. frainetto* and *Q. pubescens* were collected in Central Italy and Northeastern Greece (Supplementary Material Table S1). Seeds from all species were shipped by air mail to commercial tree nurseries in Germany, Italy and Greece, and sown in plastic trays (Quickpot 15T/16 or 24/18). In each country, the trays were filled with different substrate according to the

regional practices in order to achieve optimum growth of oaks. Specifically, in Germany trays were filled with a mixture of humus, volcanic lapilli and different organic fibres (FE Typ Bio Blumenerde # 03010, HAWITA Co., Vechta, Germany), in Italy trays were filled with a 40/40/20 mixture of peat, loam and volcanic lapilli along with $25 \text{ g} \cdot \text{L}^{-1}$ pelleted humified manure and $2 \text{ g L}^{-1}$ Osmocote Exact fertilizer, and in Greece trays were filled with a 70/30 mixture of peat and black peat along with $12 \text{ kg m}^{-3}$ clay. In Italy and Greece, the emerged seedlings overwintered outdoors, while in Germany they overwintered in a non-heated, but frost-free glasshouse. These procedures are common practice for the first winter with pot-seeded material in the different countries (in Germany, other frost-protection strategies are also applied). In Germany, 1-year-old *Q. robur* seedlings (provenance 81707 "Oberrheingraben", supplied by Darmstädter Forstbaumschulen, Darmstadt, Germany) were added to the ensemble in spring 2016. During 2016, all seedlings were grown outdoors with regular watering, while in Germany seedlings were grown under nettings in autumn. In autumn 2016, emerged seedlings were measured and scored with respect to growth and physiological parameters. In spring 2017, 420 (Germany and Greece) and 210 (Italy) seedlings per species and provenance with the highest score (cf. paragraph "Plant fitness parameter scoring and ranking" below) were transferred to the plantation sites. In each site, 210 seedlings per species and provenance were established, except for Italian *Q. frainetto* in the Greek plantations, where only 170 seedlings were available due to lower seedling emergence rates. In Greece, seedlings were planted on 24 February–4 March 2017, while in Italy and Germany seedlings were planted between 15 April and 1 May 2017.

### 2.2. Common Garden Experiments

Two plantations were established in Germany, two in Greece, and one in Italy. In Germany, one plantation site is located in a managed oak and pine forest, on fluvial sand, with about 2 m deep groundwater table, in Schwanheim, Frankfurt ("SWA", 50°04′12.6″ N, 08°33′42.2″ E, 114 m a.s.l.). SWA is a typical oak site where oak forests exist for at least 500 years. The other plantation is in Riedberg, Frankfurt ("RIE", 50°10′12.8″ N, 08°37′53.2″ E, 130 m a.s.l.) on loamy soil, where planted seedlings were irrigated when necessary, as defined by monitoring weather conditions and potential drought symptoms on the trees. RIE is a cleared agricultural field, natural vegetation here would be (mixed) beech forest. In Greece, one plantation was established in Olympiada, Chalkidiki ("OLY", 40°36′33.6″ N, 23°45′05.0″ E, 48 m a.s.l.) in natural *Q. ilex* and *Q. pubescens* forest stands with loamy soil, where water was solely provided by precipitation. The particular location of the OLY plantation is a former grazing site. The other plantation is in Stratoniki, Chalkidiki ("STR", 40°31′05.5″ N, 23°47′19.2″ E, 248 m a.s.l.) in a mixed forest in the *Q. pubescens* zone, with loamy soil, where water was also solely provided by precipitation. STR is a cleared forest site. In Italy, the plantation is established at Sant' Anatolia di Narco, Umbria, in the Central Apennine ("SAN", 42°44′18.4″ N, 12°50′17.6″ E, 290 m a.s.l.), surrounded by mixed forest including *Q. ilex* and *Q. pubescens*, on calcareous soil. SAN is a former tree nursery site. During 2017, the seedlings here were also irrigated when necessary. Moreover, weed growth was intensive due to precipitations, thus the sites were cleared 2 to 3 times per year.

Plantation schemes in the common gardens are shown in supplementary Figure S1. In each plantation, 10 plots per species and provenance were established consisting of 21 seedlings, in a 3/5/5/5/3 arrangement called "grouped" scheme (Figure S1D in supplementary material) [15–17]. In the case of Italian *Q. frainetto* in the Greek plantation, only 17 seedlings per plot were established due to the abovementioned limitation of good seedlings. This particular scheme is well suited for enhancing tree quality in oaks, as well as more economic compared to row plantation [15]. To avoid clines in the plantations, the selected trees from the evaluation/scoring procedure were distributed randomly across the plantation in a RCBD design. Except for SAN, the original individual tree IDs from the evaluation procedure in the nursery were recorded for each tree during the planting proce-

dure, thus allowing comparison of evaluation scoring data from 2016 with data measured on trees in later years.

### 2.3. Climate Data

At RIE and SWA, climate data were recorded on-site with iMetos sm SMT280 weather stations (Pessl Instruments, Weiz, Austria). For SAN, data in Val Nerina, Casteldilago (coordinates 42°34′58.8″ N, 12°46′30″ E at 270 m a.s.l.) were recorded with a Davis Vantage Pro2 weather station (Davis Wetterstationen, Neumarkt, Germany). For OLY and STR, data were recorded at the closest by weather stations of Hellas Gold SA, at 40°36′09.45″ N, 23°45′01.39″ E, 28 m.a.s.l. and at 40°31′03.42″ N, 23°47′47.96″ E, 208 m a.s.l., respectively. Due to technical problems with recording of $T_{max}$ at STR, the $T_{max}$ values from the closest weather station at Arnaia (ARN: 40°29′30.88″ N, 23°36′08.49″ E, 565 m a.s.l.) were used in the STR/ARN diagram in Figure 1. From DOY 146 on in 2017, additional data were collected in the OLY and STR sites with (unshaded) HOBO onset 8K data loggers placed at ca. 1.5 m above ground to address the real heat stress which the seedlings experienced. Precipitation data reported for OLY and STR were all taken from the Hellas Gold weather stations.

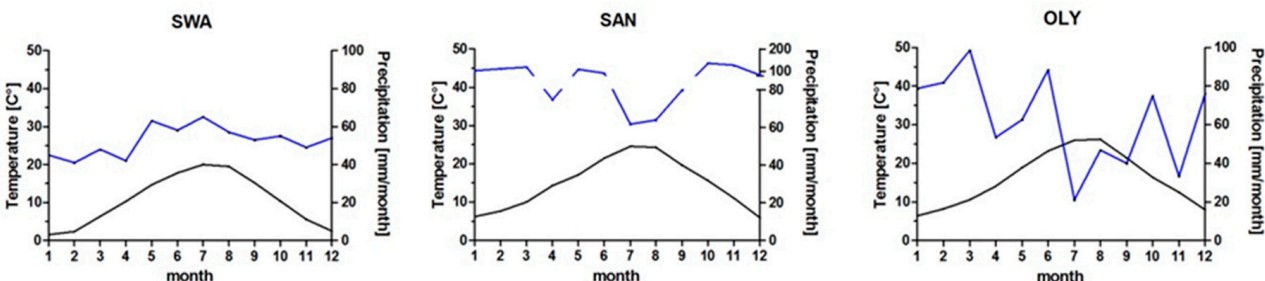

**Figure 1.** Climate diagrams for SWA, SAN and OLY. Data were averaged for 1981–2010 (SWA: Frankfurt Airport, climate station 1420 of DWD (2018)), 2012–2018 (SAN) and 2008–2017 (OLY), respectively. Black lines represent the average temperature, while blue lines represent the average precipitation.

### 2.4. Soil Analysis

5–10 representative samples of 200 mL soil each, of the top 20 cm after removal of the top organic layer were randomly collected from various parts of the respective site. Soil pH was measured in 10 mM $CaCl_2$ extracts according to DIN 19 865, part 1 (1977). Granulometric distribution was measured according to DIN 19 683 part 1 and 2 (1973). After rattle sieving with 2 mm sieves, fine soil was subjected to $H_2O_2$ treatment (for destruction of organic matter) and dispersed in 0.4 N $Na_4P_2O_7$. After wet filtration for particle sizes of coarse (2000–630 μm), medium (630–200 μm) and fine sand (200–63 μm), smaller fractions were analyzed by sedimentation analysis according to Köhn [18]. Inorganic and organic (humus) carbon contents were analyzed according to DIN 19 684, part 2 (1977) with gas analyzers LECO RC-412 and LECO EC-12 (for acidic samples) from LECO Instruments (Mönchengladbach, Germany), respectively. Organic matter was calculated from carbon content of the organic fraction by multiplication with a factor of 1.72 [19]. Initial water contents and field capacity were determined gravimetrically in fresh and water saturated samples, respectively, before and after drying for three days at 105 °C [20].

### 2.5. Morphological Attributes

In summers of 2016, 2017, 2018 and 2019, seedling heights ("*h*" in the formulas given below) were measured with rulers from soil surface to the top of the longest stem (branch), i.e., in general to the top bud. Root collar diameter was measured with an electronic Vernier caliper at 5 mm above soil level. In 2016, also the numbers of emerged leaves per seedling ("*leaf no*") were counted for fitness determination.

### 2.6. Relative Chlorophyll Content and Chlorophyll Fluorescence

In general, relative chlorophyll contents were measured in summers of 2016, 2017, 2018 and 2019 with a SPAD 502 Plus chlorophyll meter (Konica Minolta Sensing Co., Munich, Germany) on vital, fully developed leaves emerged in the respective year (important for *Q. ilex*) and given as dimensionless numbers ("*SPAD*" in the formulas below). For comparison with data obtained in Greece in 2016 with an OPTI-Sciences (Hudson, NH, USA) CCM-200 chlorophyll meter, a comparative measuring row of 152 leaves from all oak species covering the observed range of values was performed and used for determination of a calibration coefficient (to obtain "calculated SPAD values" from the CCM-200 values).

Performance index (*PI$_{abs}$* in the formulas below) is a parameter summarizing the effects of light trapping, quantum efficiency of reduction of $Q_A$ and efficiency of electron transport from $Q_{A-}$ to the intersystem carriers of the electron chain [21,22]. *PI$_{abs}$* was determined in mid- to late summer, i.e., after full leaf development with a Pocket PEA chlorophyll fluorometer (Hansatech, King's Lynn, UK) either at night or in the early morning after dawn, on leaves pre-darkened for a minimum of 20 min with leaf clips, and data were analyzed with PEA Plus 1.0.0.1 (Hansatech) software. A detailed analysis of the results of these analyses is provided in a separate paper [2].

### 2.7. Plant Fitness Parameter Scoring and Ranking

To compare plant fitness, the morphological and physiological parameters of the emerged seedlings in summer 2016 were used to perform a ranking of the individual seedlings for selection of the best-suited individuals for planting. For this purpose, an arbitrary fitness indicator (*FI*) was calculated for each individual seedling *i* according to (1):

$$FI = 0.25 * \frac{h_i}{h_{max}} + 0.25 * \frac{leaf\ no_i}{leaf\ no_{max}} + 0.25 * \frac{SPAD_i}{SPAD_{max}} + 0.25 * \frac{PIabs_i}{PIabs_{max}} \tag{1}$$

with "i" indicating each individual seedling and "max" indicating the highest value observed for this parameter in the respective population (species/provenance). For the plantations, the 420 (Italy: 210) best performing seedlings were then selected. In Greece, for *Q. frainetto* from Italy only 340 seedlings were available due to low germination rates.

Each individual tree was labeled and could be traced over time in the common garden plantations. To assess the potential predictive value of the 2016 *FI* values for seedling survival and development, we grouped the individual seedlings into "survivors" and "non-survivors" (in 2017) and compared their respective 2016 *FI* values by t-test. Furthermore, within the surviving seedlings, the rank of each individual obtained in 2016 was compared to its rank in 2019 by Spearman's correlation analysis. The 2019 ranking values of the individual were obtained omitting leaf number and SPAD, thus weighing height and *PI$_{abs}$* with a factor of 0.5 each. To compare population (species/provenance) performance at the different sites, we took survival rates into account. We calculated, for each site (or, in the case of Germany, for both sites combined, see Table 1), mean values of individual parameters ("x") for each population k (i.e., species/provenance), ("x$_k$"), then identified the maximum mean values of all population ("x$_{max}$") and calculated an arbitrary Relative Population Fitness (*RPF*) score (for each site) according to (2):

$$RPF = 0.40 * \frac{survival_k}{survival_{max}} + 0.30 * \frac{h_k}{h_{max}} + 0.30 * \frac{PIabs_k}{PIabs_{max}} \tag{2}$$

**Table 1.** Relative fitness of populations (*RPF*, cf. Equation (2)) in the different countries and common garden sites. Data were obtained in summer 2019.

| Species | Origin | Site | Survival | RPF | Ranking |
|---|---|---|---|---|---|
| German sites | | | | | |
| *Q. pubescens* | IT | RIE | 98.57 | 0.869 | 1 |
| *Q. robur* | DE | RIE | 88.10 | 0.808 | 2 |
| *Q. pubescens* | GR | RIE | 96.67 | 0.804 | 3 |
| *Q. ilex* | IT | RIE | 95.71 | 0.777 | 4 |
| *Q. pubescens* | GR | SWA | 78.57 | 0.763 | 5 |
| *Q. ilex* | GR | RIE | 94.76 | 0.740 | 6 |
| *Q. pubescens* | IT | SWA | 85.24 | 0.728 | 7 |
| *Q. robur* | DE | SWA | 81.90 | 0.690 | 8 |
| *Q. ilex* | IT | SWA | 80.48 | 0.682 | 9 |
| *Q. frainetto* | GR | RIE | 97.62 | 0.634 | 10 |
| *Q. ilex* | GR | SWA | 82.86 | 0.630 | 11 |
| *Q. frainetto* | IT | RIE | 95.71 | 0.623 | 12 |
| *Q. frainetto* | IT | SWA | 74.29 | 0.610 | 13 |
| *Q. frainetto* | GR | SWA | 72.38 | 0.609 | 14 |
| **Species** | **Origin** | **Site** | **Survival** | **RPF** | **Ranking** |
| Italian site | | | | | |
| *Q. pubescens* | IT | SAN | 81.90 | 0.944 | 1 |
| *Q. pubescens* | GR | SAN | 77.62 | 0.873 | 2 |
| *Q. ilex* | IT | SAN | 82.38 | 0.849 | 3 |
| *Q. ilex* | GR | SAN | 74.76 | 0.780 | 4 |
| *Q. frainetto* | GR | SAN | 60.95 | 0.540 | 5 |
| *Q. frainetto* | IT | SAN | 58.57 | 0.486 | 6 |
| **Species** | **Origin** | **Site** | **Survival** | **RPF** | **Ranking** |
| Greek site | | | | | |
| *Q. pubescens* | GR | OLY | 18.10 | 0.939 | 1 |
| *Q. pubescens* | IT | OLY | 11.43 | 0.820 | 2 |
| *Q. ilex* | IT | OLY | 2.86 | 0.509 | 3 |
| *Q. frainetto* | GR | OLY | 3.81 | 0.443 | 4 |
| *Q. frainetto* | IT | OLY | 4.29 | 0.429 | 5 |
| *Q. ilex* | GR | OLY | 1.90 | 0.314 | 6 |

In addition, the root collar diameter in 2016 was compared (by *t*-test) between the abovementioned "survivors" and "non-survivors" (in 2017) in order to check the parameter's potential as an indicator of second-year survival as reported by Tsakaldimi et al. [14]. Moreover, the potential of seedling height used alone as a predictive parameter for the German and the Greek sites was tested.

### 2.8. Statistical Analysis

Statistical analysis was conducted using PRISM 2.0 software (GraphPad Software, San Diego, CA, USA). Specifically, Kruskal-Wallis test with Dunn's post-hoc test at a significance level of *p* = 0.05 was conducted to identify potential significant differences between populations (i.e., site/species/provenance combinations).

## 3. Results

### 3.1. Soil Parameters

The soil parameters are given in Table 2. The German sites are very different in soil properties, with more calcareous and loamy conditions at RIE and an acidic, sandy soil in the recently partially cleared site SWA with its very high organic matter in the topsoil. The Italian site is typically calcareous and characterized by good water holding capacity. Together with its situation in the river valley, it provides very good conditions for seedling growth. Both Greek sites show slightly acidic, loamy soil with good water holding capacity,

but since they both are far away from recent superficial water sources, they both rely on precipitation for seedling water supply.

**Table 2.** Soil parameters in the plantation sites. Data are means ± SD of n = 5–6 determinations (except for SWA: means ± SE, n = 2).

| Site | SWA | RIE | SAN | OLY | STR |
|---|---|---|---|---|---|
| Granulometric analysis | | | | | |
| Sand (%) | 80.8 ± 0.8 | 11.6 ± 4.0 | 33.6 ± 6.5 | 55.2 ± 5.6 | 31.4 ± 4.2 |
| Silt (%) | 12.0 ± 0.3 | 64.1 ± 4.3 | 41.5 ± 3.7 | 26.3 ± 1.9 | 41.3 ± 3.5 |
| Clay (%) | 7.2 ± 0.5 | 24.2 ± 3.7 | 24.9 ± 2.9 | 18.5 ± 3.9 | 27.3 ± 4.4 |
| Soil type | Sl2 | Lu | Ls2-Ls3 | Ls4-Sl4 | Lt2 |
| pH (CaCl$_2$) | 3.3 ± 0.3 | 7.2 ± 0.2 | 7.2 ± 0.1 | 5.4 ± 0.2 | 6.3 ± 0.6 |
| Organic matter (%) | 36.5 ± 19.5 | 3.1 ± 2.1 | 2.2 ± 0.3 | 4.7 ± 1.2 | 17.6 ± 1.6 |
| CaCO$_3$ (%) | nd | 6.1 ± 4.6 | 56.7 ± 1.8 | 0 | 0.1 ± 0.3 |
| Field capacity (Vol-%) | 20.1 ± 4.6 | 31.1 ± 7.6 | 44.3 ± 1.1 | 45.8 ± 4.0 | 50.0 ± 3.8 |

### 3.2. Climate Data and Extreme Climate Events

While the young seedlings were under controlled watering conditions during their first year of growth, the trees in SWA, OLY and STR were dependent on precipitation after planting. In RIE and SAN, watering supply was installed and used, if the weather conditions indicated drought situations. Monthly temperature and precipitation patterns at the common gardens (2017, 2018 and 2019) are shown in Table S2, a few of which are noteworthy. In January 2017, the potted seedlings were exposed to extreme frost (subzero temperature down to −10 °C for several days) before establishing the plantations in Greece. In July and August 2018 and 2019, extremely high temperatures in combination with prolonged drought were encountered by the seedlings in Germany. Especially in SWA temperatures were higher than RIE and even exceeded 40 °C for a few days.

### 3.3. Seedling Survival and Growth

Initially, 68% of seeds germinated on average and gave rise to seedlings. The survival rates in the third year after planting (2019) are given in Table 3. Seedlings grown in Germany and Italy exhibited relatively high survival rates, i.e., 59–99%. In Greece, environmental conditions were particularly challenging for the seedlings, especially for *Q. ilex* with few individuals surviving. In general, seedlings in STR had greater survival rates compared to OLY plantation. In all five plantations, *Q. pubescens* revealed the highest survival rates among the three oak species. Moreover, the drier (sandy soil) German plantation site (SWA), and the warmer and drier (lowland) Greek plantation site (OLY) exhibited higher seedling mortality compared to RIE and STR, respectively.

**Table 3.** Survival rates (%) of seedlings after three years in the five plantation sites (August 2019).

| Plantation Site | Species and Provenance | | | | | |
|---|---|---|---|---|---|---|
| | *Q. Ilex* | | *Q. Frainetto* | | *Q. Pubescens* | |
| | Greece | Italy | Greece | Italy | Greece | Italy |
| RIE | 95 | 96 | 98 | 96 | 97 | 99 |
| SWA | 83 | 80 | 72 | 74 | 79 | 85 |
| SAN | 75 | 82 | 61 | 59 | 78 | 82 |
| OLY | 2 | 3 | 4 | 5 | 18 | 11 |
| STR | 6 | 6 | 18 | 18 | 51 | 27 |

Figure 2 depicts the initial size and the height growth of the seedlings during the first three years in the common gardens. As a general pattern, seedlings in Germany were tallest except for the Italian *Q. ilex*, which showed the highest initial size in the Italian

nursery (Gubbio, Italy). Statistical analyses after three years in the plantation sites revealed that seedling height (Table S3) was greatest for *Q. ilex* and Italian *Q. pubescens* in RIE and SAN, while *Q. frainetto* seedlings were taller mainly in the German plantations (RIE and SWA) and secondarily in SAN. In the Greek plantations, seedling height of all species was significantly lower showing practically no height growth.

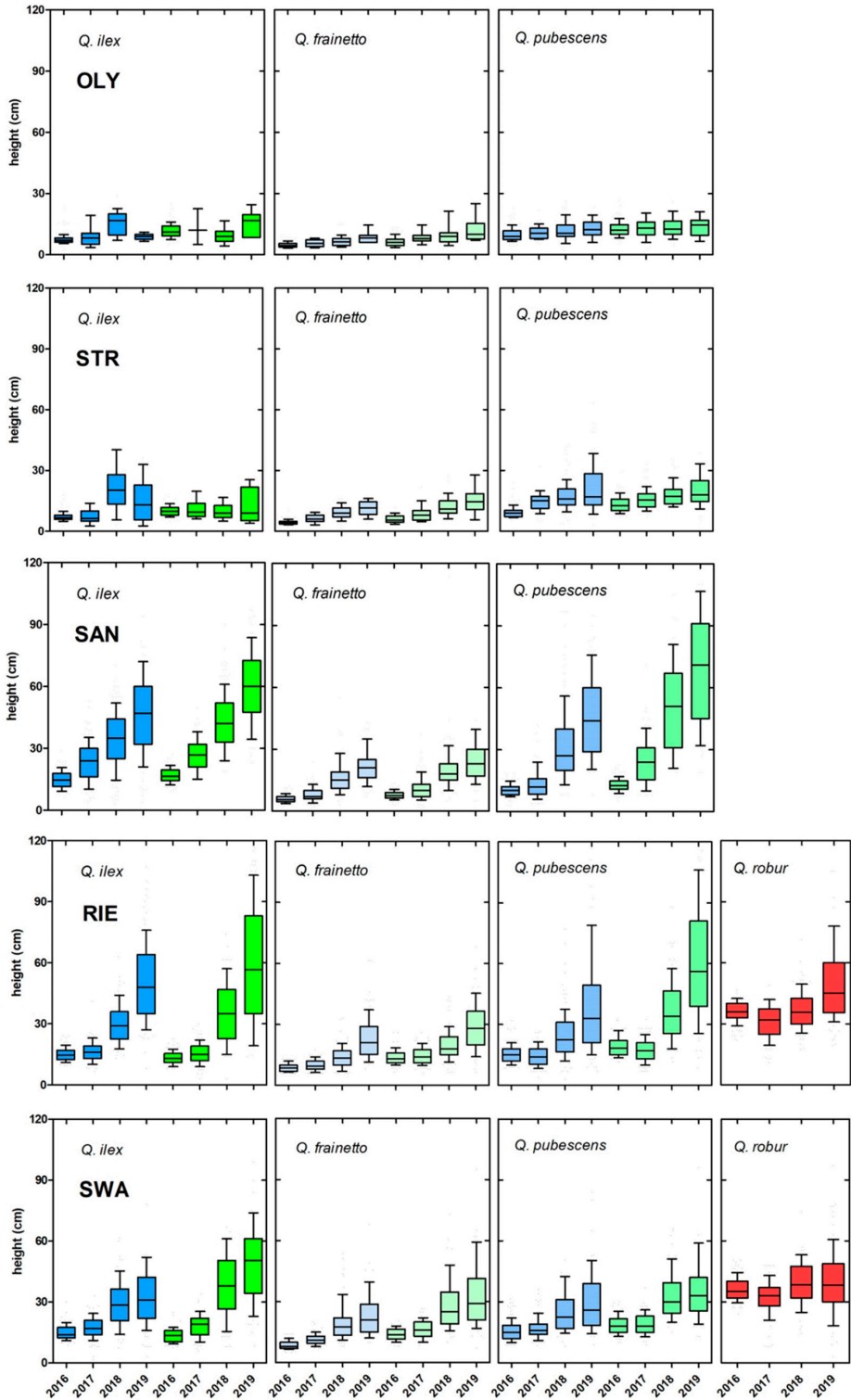

**Figure 2.** Development of seedling height in the different plantations sites. Greek provenances: blue, Italian provenances: green, *Q. robur*: red. Box plots with 10–90% whiskers.

In general, root collar diameter followed the same trend with height. Height × root collar diameter correlation analysis in 2019 revealed moderately positive (Greek *Q. frainetto* and both *Q. pubescens*) to strong positive (Italian *Q. frainetto* and both *Q. ilex*) relationships between the two parameters (Figure 3).

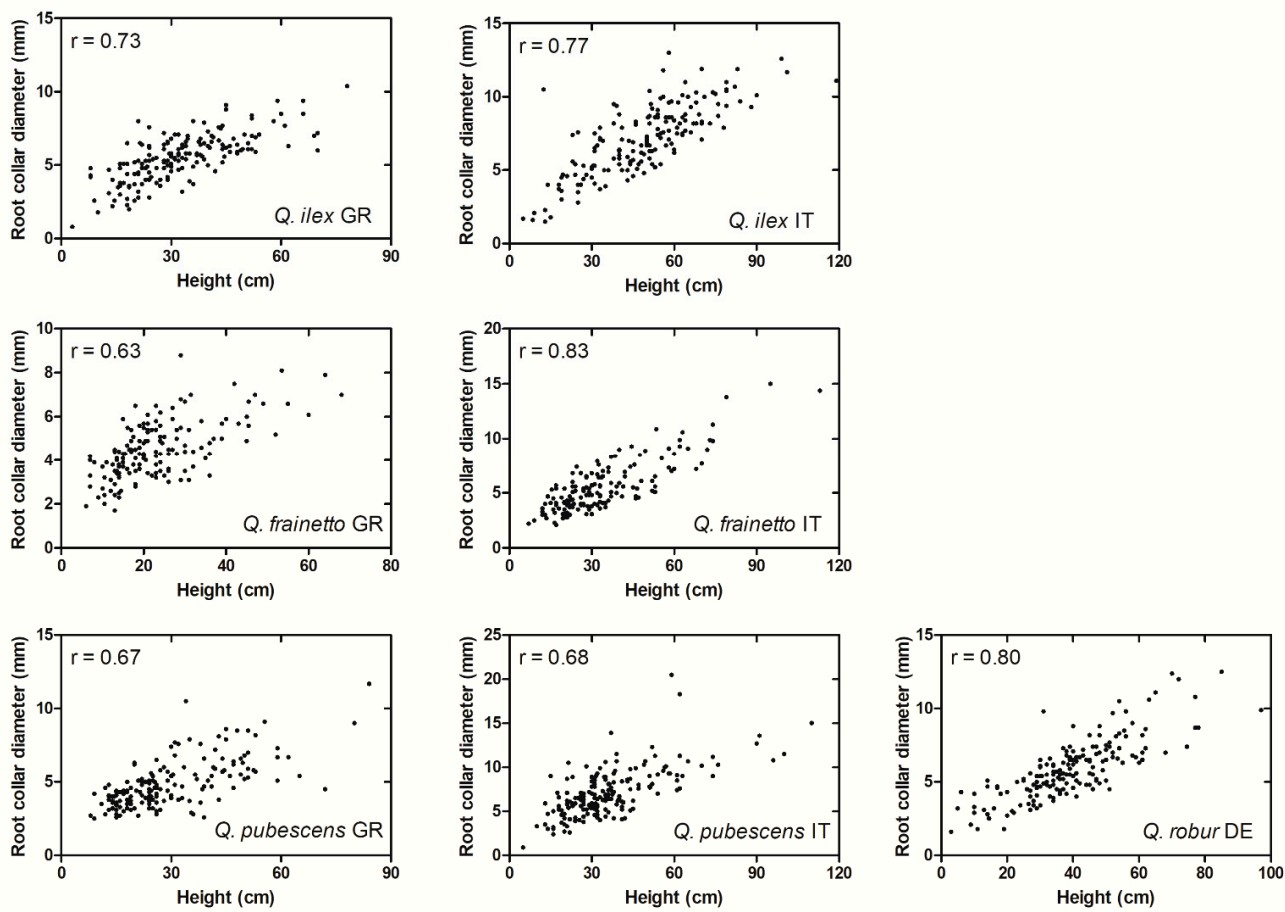

**Figure 3.** Pearson's correlation analyses between seedling height and root collar diameter in 2019 for every species × provenances tested. The "r" value represents Pearson's correlation coefficient. In all species × provenances, *p*-value was <0.0001.

### 3.4. Relative Chlorophyll Content

Table 4 depicts the SPAD values of each species established in the common gardens in the nursery and during three years after planting. *Q. ilex* provenances in general showed the lowest SPAD values in all years in STR and high values in SAN and the German sites. SPAD values of both *Q. frainetto* provenances were generally greater in SWA compared to the rest of the plantation sites. Moreover, both Greek and Italian *Q. pubescens* revealed significantly greater SPAD values mainly in SWA and secondarily in SAN. In general, SPAD values in SWA and RIE showed a tendency to increase after establishment in the field, whereas seedlings at SAN revealed the opposite response.

**Table 4.** Relative chlorophyll content (SPAD) of each species × provenance from the nursery (2016) until after three years in the different plantations sites (2017–2019). Mean values (± SE) of the different plantation sites for each species/provenance within a line for each year followed by different letters are significantly different (*P* < 5%).

| Species and Provenance | Year | Plantation Site | | | | | | | | | |
|---|---|---|---|---|---|---|---|---|---|---|---|
| | | RIE | | SWA | | SAN | | OLY | | STR | |
| *Q. ilex* GR | 2016 | 43.49 ± 0.29 | a | 43.24 ± 0.30 | a | 33.69 ± 0.55 | b | 26.10 ± 0.55 | c | 27.19 ± 0.51 | c |
| | 2017 | 36.83 ± 0.47 | b | 35.99 ± 0.39 | bc | 39.74 ± 0.37 | a | 39.00 ± 1.68 | ab | 31.70 ± 1.82 | c |
| | 2018 | 36.72 ± 0.43 | a | 37.33 ± 0.35 | a | 38.60 ± 0.57 | a | 30.71 ± 1.03 | b | 30.64 ± 1.39 | b |
| | 2019 | 33.10 ± 0.33 | a | 33.84 ± 0.42 | a | | | | | 15.77 ± 5.49 | b |
| *Q. ilex* IT | 2016 | 43.32 ± 0.33 | a | 42.88 ± 0.32 | a | 35.97 ± 0.65 | b | 33.56 ± 0.54 | c | 30.36 ± 0.52 | d |
| | 2017 | 35.49 ± 0.44 | b | 36.30 ± 0.40 | b | 41.18 ± 0.41 | a | 34.73 ± 3.89 | ab | 34.24 ± 1.97 | b |
| | 2018 | 39.14 ± 0.49 | a | 38.59 ± 0.30 | a | 40.19 ± 0.52 | a | 30.97 ± 2.45 | b | 29.06 ± 1.65 | b |
| | 2019 | 33.95 ± 0.41 | a | 34.81 ± 0.38 | a | | | | | 22.13 ± 4.40 | b |
| *Q. frainetto* GR | 2016 | 38.76 ± 0.31 | a | 38.34 ± 0.33 | a | 18.09 ± 0.29 | c | 24.72 ± 0.63 | b | 25.22 ± 0.63 | b |
| | 2017 | 33.66 ± 0.47 | b | 37.10 ± 0.49 | a | 30.28 ± 0.56 | c | 30.98 ± 1.28 | bc | 30.05 ± 0.96 | c |
| | 2018 | 33.51 ± 0.38 | b | 37.08 ± 0.31 | a | 24.48 ± 0.89 | d | 23.75 ± 1.06 | d | 28.08 ± 0.80 | c |
| | 2019 | 27.51 ± 0.51 | b | 35.74 ± 0.34 | a | | | | | 19.93 ± 1.01 | c |
| *Q. frainetto* IT | 2016 | 40.84 ± 0.27 | a | 41.42 ± 0.26 | a | 19.47 ± 0.49 | c | 23.52 ± 0.59 | b | 23.82 ± 0.60 | b |
| | 2017 | 32.01 ± 0.42 | b | 36.61 ± 0.45 | a | 32.54 ± 0.48 | b | 30.88 ± 1.02 | b | 32.71 ± 1.06 | b |
| | 2018 | 30.85 ± 0.41 | b | 36.44 ± 0.32 | a | 24.48 ± 0.82 | c | 22.23 ± 0.88 | c | 31.06 ± 1.19 | b |
| | 2019 | 25.97 ± 0.48 | b | 34.72 ± 0.34 | a | | | | | 16.70 ± 1.47 | c |
| *Q. pubescens* GR | 2016 | 41.85 ± 0.30 | a | 41.93 ± 0.30 | a | 29.29 ± 0.54 | c | 33.65 ± 0.55 | b | 35.25 ± 0.53 | b |
| | 2017 | 37.05 ± 0.40 | b | 40.16 ± 0.36 | a | 40.45 ± 0.39 | a | 33.56 ± 0.60 | c | 34.54 ± 0.69 | c |
| | 2018 | 37.79 ± 0.32 | b | 40.18 ± 0.23 | a | 39.15 ± 0.49 | ab | 28.90 ± 0.76 | c | 31.64 ± 0.59 | c |
| | 2019 | 36.23 ± 0.28 | b | 38.48 ± 0.25 | a | | | | | 29.07 ± 1.32 | c |
| *Q. pubescens* IT | 2016 | 40.57 ± 0.24 | a | 40.76 ± 0.26 | a | 27.25 ± 0.46 | b | 28.52 ± 0.51 | b | 27.37 ± 0.50 | b |
| | 2017 | 36.08 ± 0.41 | b | 38.15 ± 0.39 | a | 39.20 ± 0.40 | a | 32.99 ± 0.88 | c | 32.19 ± 0.83 | c |
| | 2018 | 35.19 ± 0.32 | c | 39.66 ± 0.25 | a | 37.50 ± 0.47 | b | 23.59 ± 0.84 | e | 28.43 ± 0.68 | d |
| | 2019 | 34.77 ± 0.32 | b | 37.53 ± 0.36 | a | | | | | 19.83 ± 1.16 | c |
| *Q. robur* DE | 2016 | 37.54 ± 0.41 | a | 37.89 ± 0.36 | a | | | | | | |
| | 2017 | 27.73 ± 0.51 | b | 36.14 ± 0.57 | a | | | | | | |
| | 2018 | 33.25 ± 0.46 | b | 36.10 ± 0.42 | a | | | | | | |
| | 2019 | 32.39 ± 0.45 | b | 37.23 ± 0.44 | a | | | | | | |

*3.5. Quality Scoring of Seedlings as Predictors for Survival and Growth and Population Fitness*

To analyze, whether root collar diameter measured in the first year to predict second-year survival of oak seedlings, the length data from 2016 of the perished (2017) and the live seedlings in 2017 were arranged into separate groups for all seedlings planted in Germany (favourable conditions) and in Greece (unfavourable conditions), respectively, and yielded statistically significant differences were tested, as suggested by Tsakaldimi et al. [14]. No significant differences were observed between survived (2017) and perished seedlings in SWA and STR. However, survived *Q. frainetto* from both provenances and *Q. pubescens* IT in RIE, and *Q. frainetto* IT in OLY developed significantly thicker stems (in 2016) compared to the respective perished seedlings (Supplemental Material Table S4). To further elucidate potential differences between the survivors and the perished trees at the sites, where stem diameter alone revealed no indicative value, a more complex parameter was also developed, taking into account both morphological and physiological fitness (i.e., *FI*, a function of tree length, leaf development, chlorophyll content and efficiency of the photosynthetic apparatus (Equation (1)). However, in no combination of provenance and country of plantation the 2016 *FI* values of the subpopulations (live in 2017 versus dead in 2017) yielded significant differences, meaning that the 2016 *FI* values could not be used as predictors for seedling survival (Supplemental Material Table S5).

In addition, the relationship between the *FI* values of individual trees in 2016 and 2019, respectively, was studied, for the German plantations to assess the predictive value of seedling measurements for future growth (Supplemental Material Table S6). In the German plantations, where 72–99% of the seedlings survived into 2019 in all provenances, the Spearman correlation coefficients between the *RPF* in 2016 and 2019 were weak (i.e., <0.21) and calculated *RPF* obtained in 2016 were considered of little predictive value for the future performance of individual seedlings. If data for seedling height in 2016 were compared to those of 2019, in all cases weak correlations were found both in the German and in the Greek plantations (Pearson's r < 0.3).

In Table 1, all populations per site have been evaluated for their relative vitality, using survival rates, morphometric and physiological data obtained in 2019. Data for STR had to be omitted due to road blocking by storm-felled trees in 2019 which prevented accessibility at night for $PI_{abs}$ measurements. However, the site was accessible at daytime and data for survival and height at STR are given in Table 3 and Figure 2, respectively. The best performing population in Germany was the Italian *Q. pubescens* followed by *Q. robur* and the Greek *Q. pubescens*, all in RIE, while *Q. frainetto* showed the lowest *RPF*. In Germany, RIE populations showed higher vitality scores compared to SWA. In Italy, *Q. pubescens* showed the best overall performance with *Q. ilex* following. In Greece, *Q. pubescens* showed relatively higher vitality scores compared to the other oak species.

## 4. Discussion and Conclusions

The present article provides an early evaluation of three oak species' establishment in four common garden studies, including nursery stage as well as field performance during the first three years after field plantation. The seedling production process followed the protocols of local nurseries. It was decided to not impose a standardized procedure on the nurseries, since each of them had its own year-long experience with the local oak provenances under the local climatic conditions. The same argument holds for the different plantation procedures. It was not the aim of the study to experiment with the local standardized growth techniques, since this would have introduced additional technical problems in procedure (e.g., shipping of plantation soil from one country to another, missing availability of frost-free greenhouses in Gubbio and Olympiada) and site management. The authors were aware that the different treatments at different sites would prevent a more comprehensive comparison of seedling performance with solely the different macroclimates (and soil conditions) as variables. However, the cooperation with the local nurseries and their local experiences with the growth of oaks reflect the situation in real-life transfer of genotypes from one country to the other.

The potential to include a German population of *Q. pubescens* was also examined. In German nurseries, no *Q. pubescens* seeds or seedlings are available from German populations, since these are mostly very small, often include hybrids with the local species, and, to our knowledge, are thus usually not included in certified seedling material lists. It was decided against self-collecting such seeds due to the genetic uncertainties as well as due to the fact that it was not possible to predict if there would be sufficient seeds available in 2016. Since the aim of the study was not translocating German material to the Mediterranean countries, the German control material was confined to the local species at the plantation sites (i.e., *Q. robur*).

### 4.1. Seedling Survival and Establishment

Seedling emergence was good for all collected seed populations, giving rise to sufficient plant material for an initial selection step, except for *Q. frainetto* of Italian origin at the Greek nursery. Transportation of the seeds in perforated plastic bags by airmail may have led to partial anoxia in the center of the bags, thus reducing germination rates, but in general, the outcome of seedlings was satisfactory. After their respective plantation in winter 2016/17, survival rates were assessed in summer 2019, and except for the Greek plantations, seedlings had well established themselves in the Common Gardens (Table 3). Initially it was intended to irrigate the seedlings in one of the Greek sites but this ultimately turned out to be impossible due to technical problems, i.e., the large distance in the finally available plantation site from available water. At RIE and SAN, facilities for watering of the seedlings had been established beforehand, so during severe soil drought in summer, these seedlings could be watered. Here, as well as in SWA, where no watering occurred, survival rates of all seedlings were satisfactory. In Greece, seedlings encountered a period of extreme frost in early January 2017 before establishing the plantations, and the evergreen *Q. ilex* was severely affected. Low seedling vitality has also been described for other broadleaves after freezing [23] and in summer-dry, calcareous Mediterranean habitats (e.g., Pausas et al. [12]: only ca. 15% survival of *Q. ilex* ssp. *ballota* after 3 years). During planting at the beginning of March, the seedlings still looked alive with most of the leaves still green, but turned out to be lethally damaged when the growth period started. For *Q. ilex* from its northernmost population in Italy, North of Lake Garda, it is known that frost tolerance (50% survival) of 1-year-old seedlings is around −10 (leaves) to −15 °C (sprout cambium) [24]. Such temperatures have, to our knowledge, never been recorded in the area of Olympiada, where the seedling material stems from, nor in Latium, Italy, close to the sea, where the Italian seeds have been collected. Since temperatures at soil level, where the seedlings overwintered outdoors, may also have been well below the −9.7 °C recorded at the Olympiada weather station (i.e., at 2 m height), we therefore conclude, that stems, roots and leaves of most *Q. ilex* seedlings were irreversibly damaged already before the planting. In contrast to the seedlings, the predominant mature *Q. ilex/Q. pubescens* forest in the surroundings of the plantation at OLY was not visibly affected by the frost event (unpublished field observations in May 2017), in accordance with the findings at Lake Garda, where adult trees revealed frost tolerance of −25 °C (stem cambium, 50% survival, [24]). The deciduous species, *Q. frainetto* and *Q. pubescens*, which occur naturally at higher elevations in Chalkidiki, northern Greece, obviously developed better frost tolerance of stem and cambium (and even roots). Although *Q. frainetto* seedling survival rates in the field are better when planted earlier and allowed to develop a better root system (i.e., Dec/Jan, [25]), in the current experiment we opted for a later planting date because of the severe frost in January 2017. However, in March/early April 2017 precipitation was low, especially at OLY (Table S2), and many seedlings died of drought due to weakly developed root systems before sufficient rainfalls occurred in mid-April. *Q. pubescens* showed better survival rates compared to *Q. frainetto* under identical conditions (Table 1) confirming similar findings in Northern Greece [26]. *Q. frainetto* has been found to be the least drought-tolerant oak among four Mediterranean species including *Q. pubescens* and *Q. ilex* [27], and this may explain its low survival rate at OLY and STR after the spring drought

immediately after plantation. At SAN and RIE, in contrast, seedlings were watered when necessary and therefore mortality was much lower. In SWA, although seedlings relied on precipitation, mortality was also low. Here, strong rainfalls occurred in mid-April 2017 before the flushing of the trees, which is usually observed one month later in Germany than in Greece.

Concerning the predictive value of morphological and physiological properties (chl content, $PI_{abs}$) of 1-year old seedlings for their future survival and development in the field, our results did not meet our expectations. Bayala et al. [28], on five tropical tree species, and Tsakaldimi et al. [14] on five Mediterranean tree species, including *Q. ilex* and *Q. coccifera*, pointed to stem diameter as a good indicator for second-year survival after outplanting. We examined this proposed indicator and the results were variable among the common gardens. In RIE, where seedlings showed the greatest survival, two-year old seedlings of *Q. frainetto* (both provenances) and *Q. pubescens* IT were successfully grouped to survived or perished depending on their root collar diameter, thus confirming the results of Tsakaldimi et al. [14] (Table S4). In general, root collar diameter of *Q. frainetto* showed a greater potential to predict second-year survival but differences were not always significant. While the morphological features height and root collar diameter were well correlated in older seedlings, confirming data of Pausas et al. [12] on *Q. ilex* ssp. *ballota* (Figure 2), the *FI* values in the subgroups of established (survivors into 2017) and perished seedlings (dead in 2017) did not differ significantly (Table S5). Secondly, the Spearman r values between the *FI* values in 2016 (using seedling height, number of leaves, chl content and $PI_{abs}$) and 2019 (using seedling height and $PI_{abs}$) were considered too low to be of practical value (cf. Supplementary Material Table S6: r was either not significant or even negative in all but one cases). Quite similarly, if measurements are confined to height the predictive value is also weak, with Pearson coefficients for the respective 2016 and 2019 values being negative or below 0.17 in all cases, thus height alone is neither considered suitable for preselection for plantation purposes.

In conclusion, of the parameters tested, only in good growing conditions (i.e., RIE), root collar diameter of 2nd year seedlings had a moderate predictive value for seedling survival in three out of seven species/provenances, while other non-destructive parameters (included in the *FI* values) appear to be of no practical value.

### 4.2. Cumulative Growth Conditions at the Common Garden Sites as Reflected in Seedling Performance

Height and root collar growth during the first three years after planting (measured in summers 2017, 2018 and 2019) can be used as first predictors for the suitability of the respective oak provenances for establishment at the different sites. Generally, both parameters were relatively high at SAN for most species/provenances over the three years, reflecting the good nutrient conditions (cations) in the basic soil and the sufficient water supply. In the German sites, *Q. pubescens* and *Q. robur* revealed no height growth during the first year, pointing to the costs of acclimation (i.e., root development) after planting. However, in the following years the Italian *Q. pubescens* in RIE developed strongly and reached comparable height with the SAN homologue most probably due to a well-developed root system. In the first year at the German sites, *Q. frainetto* and *Q. ilex* seedlings at SWA in general showed a stronger height growth than at RIE, possibly because of the lower light availability at SWA (thinned pine umbrella plantation with 50 trees/ha [5]) as compared to the open setup at RIE. However, after three years both *Q. ilex* provenances as well as *Q. robur* were significantly promoted by the relatively favorable conditions in RIE (i.e., irrigation) compared to SWA. After three years at the Greek sites, especially *Q. pubescens* revealed better height growth at STR than at OLY, again possibly due to partial shading (cf. Supplementary Material Table S3).

The quite consistent higher chl contents of *Q. frainetto*, *Q. pubescens* and *Q. robur* at SWA vs. RIE can be explained by partial shading by pine trees at SWA, whereas seedlings at RIE grew in full sunlight. At SAN, seedlings were also partially shaded by weeds during the period of measurements which caused early senescence of *Q. frainetto* and subsequently

low chl content values. Leaf senescence depends on the carbon fixation efficiency which is modulated by plant photoreceptors such as phytochrome A [29]. STR and OLY seedlings generally showed similar (low) chl content over the three-year period as compared to the other sites. At STR, the high irradiation, at least compared to SWA and RIE, may have caused the lower chl contents. Additionally, OLY seedlings were exposed to full sunlight, combined with high temperature and a long drought period. This combination accelerated photoinhibition [2]. Frequent photoinhibition, in turn, may cause photodamage to pigments. Similarly, Cotrozzi et al. [30] reported leaf injuries (i.e., yellowing) induced by drought stress in seedlings of *Q. ilex*, *Q. pubescens* and *Q. cerris*. In the German plantations, lower SPAD values were observed in 2017, 2018 and 2019 (outdoor grown seedlings) vs. data from 2016 (greenhouse-grown seedlings).

The advantage which the seedlings in Germany had by initial greenhouse growth over the Italian seedlings at the time of planting (compare seedling heights in 2016 in Figure 2) was (mostly) lost in the course of outdoor growth in 2019. This is partly due to prolonged extreme heat and drought periods in 2018 and 2019 (Table S2) which showed similar intensity with the heat and drought in Europe in 2003 [31]. The most severe growth conditions existed at the Greek sites, nearly extinguishing the *Q. ilex* populations by the very unusual extreme frost in January 2017 and affecting growth in the deciduous species e.g., by spring drought.

*4.3. Preliminary Conclusions for (Re)forestation Strategies*

In summary, we can conclude that:

(1) Of the Mediterranean species, *Q. pubescens* shows the best performance at all sites, followed by *Q. ilex*. *Q. frainetto* showed the lowest performance under all conditions, except for Greek *Q. ilex* at OLY. Since this finding was independent of provenance and site, it appears to be an inherent trait and we conclude that among the deciduous species, *Q. frainetto* is a "slow-growing" species (sensu Lambers and Poorter, [32]) as opposite to *Q. pubescens* (and *Q. robur*).

(2) Whether Greek or Italian provenances perform better at a given site depends on site conditions. In general, the differences between the provenances were small.

(3) In Germany, *Q. robur* did not outperform *Q. pubescens*. Both sites (RIE and SWA) were quite warm and suffered from heatwaves during the experimental period, which may have masked a potential advantage (under milder temperatures and higher precipitation) of the local species over the Mediterranean one.

(4) The predictive value of the morphological and physiological measurements on the seedlings in 2016 is not considered of practical use, except for root collar diameter in certain combinations of provenance and growth site.

Presently, differences between Italian and Greek populations are small, and it will be necessary to monitor seedling development for more years to decide, whether the Italian populations show better performance in the long run. At OLY and STR, growth conditions are extremely hard for young oaks. To regenerate *Q. ilex* forest (especially at the lower, dryer site OLY), several attempts in successive years may be necessary to avoid frost damage, if extreme situations like in January 2017 are to occur in future years again. It is further recommended to irrigate new plantations of rooted seedlings in this region during the first one or two years [33]. In Germany, *Q. pubescens* and *Q. robur* showed the highest relative vitality scores (Table 1) in both sites, despite the fact that the two species showed minimal height growth during the first year after planting. *Q. frainetto* showed inferior quality compared to the other oaks. Consequently, *Q. pubescens* may be considered a potential future forest tree on poor and dry soils for the expected future summer heat and drought conditions in Central Europe, as were observed in Germany in 2018 and 2019. Whether Italian or Greek provenances are better suited for planting of Mediterranean species in southwest Germany should be addressed by future research in the common gardens.

**Supplementary Materials:** The following are available online at https://www.mdpi.com/article/10.3390/f12060678/s1, TableS1: Oak species, collection sites and nursery sites. Table S2: Average minimum and maximum monthly temperatures and total monthly precipitation recorded in the common gardens in the course of 2017, 2018 and 2019. Data in RIE, SWA were recorded on site, data in SAN at the climate station in Casteldilago, and data for OLY and STR close to the sites at the climate stations of Hellas Gold. Table S3: Statistical analysis of height (in cm) in the common gardens, measured in summer 2019. Values followed by different letters within a row are significantly different at $P < 5\%$, as calculated by Kruskal-Wallis one-way-ANOVA tests with Dunn´s post-hoc test. Means of *Q. robur* were analyzed by t-test at $P < 5\%$. Table S4: Statistical analysis of root collar diameter (mm) in the common gardens, measured in summer 2016, to examine the parameters potential as indicator of second-year survival after outplanting. Values followed by different letters within a row are significantly different at $P < 5\%$, as calculated by t-tests. Table S5: Mean values of fitness indicators (*FI*) of the plants in 2016, separated into those who established live in 2017 and those who perished in the first year after plantation (dead in 2017). Table S6: Spearman correlation analyses between Relative Population Fitness (*RPF*) ranking positions in 2019 vs. 2016 at the German sites. *RPF* were calculated as described in Materials and methods. Pearson´s r value was also calculated for the height data in 2019 vs. 2016 at the German and Greek sites. Significance values for the correlations are given at $P < 5\%$ (*), 1% (**), 0,1% (***). Figure S1: Plantation schemes at RIE, SWA, and SAN (1A), OLY (1B), and STR (1C). Within each plot, individual trees were planted according to 1D; in the case of *Q. frainetto* from Italy at OLY and STR, trees number 02, 09, 13 and 20 were omitted.

**Author Contributions:** Conceptualization, W.B.; methodology, W.B., K.R. and F.B. (Filippo Bussotti); investigation, F.B. (Filippos Bantis), J.G. and E.F.; data curation, W.B., E.F. and F.B. (Filippos Bantis); writing—original draft preparation, F.B. (Filippos Bantis) and W.B.; writing—review and editing, F.B. (Filippos Bantis), J.G., E.F., K.R., F.B. (Filippo Bussotti) and W.B.; supervision, project administration and funding acquisition, W.B. All authors have read and agreed to the published version of the manuscript.

**Funding:** This work was financed by grant No. 01DS15014 by the German Federal Ministry of Education and Research (BMBF) to W.B.

**Data Availability Statement:** The data presented in this study are available within the article and its supplementary materials.

**Acknowledgments:** Special thanks are due to the City of Frankfurt/Main (Germany), to Greek Nurseries SA (Olympiada, Greece), Hellas Gold SA (Athens, Greece) and to Agenzia Forestale Regionale (Perugia, Umbria, Italy) for providing the respective plantation sites and help with establishment and maintenance of the plantations as well as for providing climate data (OLY and STR). Thanks are due to Martina Pollastrini (Univ. Firenze), Vera Holland, Sonja Ströll, Nathalie Reininger and Lisa Schäfer (Univ. Frankfurt) for help with the data acquisition, and also to Mariangela Fotelli for commenting on the manuscript.

**Conflicts of Interest:** The authors declare no conflict of interest.

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
