# Peer review of "Field Performances of Mediterranean Oaks in Replicate Common Gardens for Future Reforestation under Climate Change in Central and Southern Europe: First Results from a Four-Year Study"

_forests, doi:10.3390/f12060678_

Round 1

Reviewer 1 Report

The manuscript presents a study on the feasibility of the use of four southern European oak species for “reforestation” purposes in a climate change scenario, based on the performances of oak seedlings. The research is conducted in three countries of central and southern Europe over three years, with the aim of detecting the suitability of different populations of different species to be used in plantations outside their native locations. Despite the width of the work and its relevant spatial and temporal coverage, I have serious concerns about its theoretical and methodological soundness.

The first point is that the claim to support natural vegetation dynamics by assisting southern trees in their migration northwards during global warming is probably unrealistic. In this context, the authors talk about “reforestation”, but there is a huge difference between a tree plantation and a forest. It is clear that they talk about tree plantations for economic exploitation throughout the manuscript, and not about forests.

Apart from these, I do not understand why the seedlings were subjected to different treatments in the various common gardens. I understand the difficulties in conducting a homogeneous experiment, but this totally compromises the scientific soundness of the work. Correctly comparing the performances of different populations of different tree species translocated in different growing sites is challenging per se; by growing them under different cultural conditions (e.g., irrigation vs no irrigation; sown in winter vs sown in spring; different soil mixture, and so on) the authors introduce a considerable amount of bias. It is not surprising that seedlings of a couple of species grown in Germany outperformed the others since they were grown under the most favorable conditions, so far that they overwintered in a frost-free greenhouse, as opposed to some of the Greek ones that were exposed to extreme frost in the same winter. As regards irrigation, the authors claim that the two irrigated sites were taken as controls due to the fact they were irrigated; if having a control was the aim, this is not correct as well, since they also claim to have irrigated them “when necessary”, which assumes no controlled and fixed amount of water was provided.

Further comments:

Line 15 and elsewhere: Why didn’t you use German Q. pubescens?

Line 37; 40: you are only referring to economic value.

Line 45: ecological consequences were never mentioned before in the text.

Lines 47-48: at least Q. rubra is an invasive species in central Europe, not feasible to be planted.

Lines 160-161: you need to briefly explain here too.

Lines 324-335: these are methods.

Figure 3: no significance value is reported for such correlations; are they statistically significant?

Lines 372-373: the same cold snap also hit Italy at the beginning of January 2017. It is surprising that only Greek seedlings had consequences.

Lines 380-383: the northernmost populations of Italian Q. ilex are for sure not exposed to such temperatures, since they are relict populations growing as refugees on calcareous cliffs, not in the valleys. Northerly winter winds falling from the Alps are warm and dry (Föhn), and cause unusually mild temperatures.

Line 468-471: I do not think you can claim this.

Author Response

The manuscript presents a study on the feasibility of the use of four southern European oak species for “reforestation” purposes in a climate change scenario, based on the performances of oak seedlings. The research is conducted in three countries of central and southern Europe over three years, with the aim of detecting the suitability of different populations of different species to be used in plantations outside their native locations. Despite the width of the work and its relevant spatial and temporal coverage, I have serious concerns about its theoretical and methodological soundness.

The first point is that the claim to support natural vegetation dynamics by assisting southern trees in their migration northwards during global warming is probably unrealistic. In this context, the authors talk about “reforestation”, but there is a huge difference between a tree plantation and a forest. It is clear that they talk about tree plantations for economic exploitation throughout the manuscript, and not about forests.

  • Response: The economic implications were removed from the introduction in order to highlight the common gardens’ attempt to investigate adaptation to climate change. The authors recognized the ecological importance of common gardens and how to provide solutions to maintain oak forests.

Apart from these, I do not understand why the seedlings were subjected to different treatments in the various common gardens. I understand the difficulties in conducting a homogeneous experiment, but this totally compromises the scientific soundness of the work. Correctly comparing the performances of different populations of different tree species translocated in different growing sites is challenging per se; by growing them under different cultural conditions (e.g., irrigation vs no irrigation; sown in winter vs sown in spring; different soil mixture, and so on) the authors introduce a considerable amount of bias. It is not surprising that seedlings of a couple of species grown in Germany outperformed the others since they were grown under the most favorable conditions, so far that they overwintered in a frost-free greenhouse, as opposed to some of the Greek ones that were exposed to extreme frost in the same winter. As regards irrigation, the authors claim that the two irrigated sites were taken as controls due to the fact they were irrigated; if having a control was the aim, this is not correct as well, since they also claim to have irrigated them “when necessary”, which assumes no controlled and fixed amount of water was provided.

  • Response: The manuscript was amended in order to qualify as a field performance report and provide an evaluation of the studied species’ establishment and growth in the common gardens during the first three years. Therefore, SAN and RIE are no longer considered as Control treatments. The seedling production process followed the protocol of local nurseries. The authors decided to not impose a standardized procedure on the nurseries, since each of them had its own year-long experience with the local oak provenances. The same argument holds for the different plantation procedures. It was not the aim of the study to experiment with the local standardized growth techniques, since this would have introduced additional technical problems in procedure (e.g. shipping of plantation soil from one country to another etc., missing availability of frost-free greenhouses in Gubbio and Olympiada) and site management. Regarding irrigation in the Greek sites, initially the authors intended to irrigate the seedlings but this was not possible due to technical problems such as blocked roads and the large distance in the finally available plantation site from available water. Irrigation at RIE and SAN occurred, whenever weather data indicated potential dry spells. Trees were then controlled and water was supplied, whenever first signs of drought occurred.

Further comments:

Line 15 and elsewhere: Why didn’t you use German Q. pubescens?

  • Response: In German nurseries, no Q. pubescens seeds or seedlings are available from German populations, since these are very few, genetically not well characterized with respect to hybridization, and not included in certified seedling material lists. The authors decided against self-collecting such seeds due to the genetic uncertainties as well as due to the fact that it was not possible to predict if there would be sufficient seeds available in 2016. Since the aim of the study was not translocating German material to the Mediterranean countries, the authors confined their German control material to the local species at the plantation sites (i.e. robur).

Line 37; 40: you are only referring to economic value.

  • Response, L41-46: This part was removed as an answer to a previous comment.

Line 45: ecological consequences were never mentioned before in the text.

  • Response, L50-53: This part was removed as an answer to a previous comment.

Lines 47-48: at least Q. rubra is an invasive species in central Europe, not feasible to be planted.

  • Response, L52-53: In Germany, rubra is a species commonly planted by foresters for reforestation purposes (despite its potential as an invasive species). Personally, the authors agree with the view of the reviewer that it should not be used for reforestation in Germany, mainly for ecological reasons (litter degradation in the field is very slow and influences on accompanying vegetation and fauna are considered negative).

Lines 160-161: you need to briefly explain here too.

  • Response, L140-177: Details about the plantation sites were included as suggested.

Lines 324-335: these are methods.

  • Response, L347-358: Appropriate parts of this paragraph were moved to the Materials and Methods section (L246-247 and L263-264), as suggested.

Figure 3: no significance value is reported for such correlations; are they statistically significant?

  • Response: p-value was added in the caption of figure 3, as suggested.

Lines 372-373: the same cold snap also hit Italy at the beginning of January 2017. It is surprising that only Greek seedlings had consequences.

  • Response, L395-397: In January 2017, seedlings in the Italian nursery (Gubbio region) suffered from about -6 oC compared to -16 oC observed in the Greek nursery (Olympiada region) (cf. https://www.worldweatheronline.com/gubbio-weather-history/emilia-romagna/it.aspx).

Lines 380-383: the northernmost populations of Italian Q. ilex are for sure not exposed to such temperatures, since they are relict populations growing as refugees on calcareous cliffs, not in the valleys. Northerly winter winds falling from the Alps are warm and dry (Föhn), and cause unusually mild temperatures.

  • Response, L404-407: The authors agree with the reviewer about the special Mediterranean climate at Lake Garda. However, Q. ilex goes northward to the valley of Lake Toblino, where it may become very cold in clear late winter nights. Anyway, this part was deleted.

Line 468-471: I do not think you can claim this.

  • Response L599-502: This phrase was removed as suggested.

Reviewer 2 Report

The manuscript describes the performance of four oak species (Quercus ilex, Q. pubescens, Q. frainetto and Q. robur) under different climate conditions. Seedlings were outplanted in five common gardens established in Central Italy (one site at 290 masl, SAN, with irrigation facilities considered as Mediterranean control site), NE Greece (two sites: one at 48 masl, OLY, and the other at 248 masl, STR, both with natural irrigation only) and Southern Germany (two sites: one site exposed to summer drought, SWA, and another one well-watered control site, RIE). The monitoring for survival rates 3 years after plantation showed low mortality in the German and the Italian sites and high mortality rates in the Greek sites triggered by extreme frost and drought events. In Greece and Italy, Q. pubescens was the best performing species, whereas in Germany, Q. pubescens and Q. robur performed best. The authors concluded that Mediterranean provenances (i.e., Greek, or Italian) might be used for future forestation objectives in Central Europe. Authors also remark that to ensure the success of seedling establishment in Quercus plantations in Northern Greece, water supply may be crucial.

To better understand the response of different oak species to changes in climate and the amount of water available, is certainly helpful to improve our predictions regarding their response to the ongoing climate change. However, I have some issues with this study that prevents me to provide a very positive response. Why the authors did not install a Mediterranean control site in Greece instead of two drought sites and a drought site in Italy paired with the installed control (SAN). As occurred in the Greek sites due to the frost and drought events, having a control site in other provenance not affected by these extreme climatic episodes don´t allow to compare between “treatments” properly.

I have some minor concerns that should be addressed and clarified. I hope that my comments can help to clarify and improve some points.

Specific comments:

I miss some lines placing the novelty of the content and highlighting the significance and of the study in the abstract section.

(L80-93) Along the introduction, when authors state the objectives of the study, in my opinion, they give too detailed information about sites and species that sounds quite redundant with 2.2 section.

(L33-161) The experimental design needs some clarification. The authors refer to a previous study for more detailed information, however, in my opinion it is complex enough to allow attaching a summary diagram of the set up.

Regarding to watering, maybe I misunderstood something, did you specify somewhere in the text the amount of water supply compared with the control? Did you supply by irrigation a specific amount simulating additional “normal” rainfall events respect to the control sites?

(L138; L141) For the international audience specifying “near…” doesn´t add valuable information. If someone is interested in the specific location, she/he can head to coordinates.

Results section: please review the verbal tense, sometimes is mixed present and past, authors must be consistent through the text.

(L256) Avoid subjective appreciations like “we can expect…” in the results section.

(L330) Change the verbal tense for consistence “scores obtained in 2016 were considered”.

I found the discussion better structured and explanatory than results section.

Being one of the main questions, concluding remarks on point 5 (predictive value of morphological and physiological measurements) are vague.  

Author Response

The manuscript describes the performance of four oak species (Quercus ilex, Q. pubescens, Q. frainetto and Q. robur) under different climate conditions. Seedlings were outplanted in five common gardens established in Central Italy (one site at 290 masl, SAN, with irrigation facilities considered as Mediterranean control site), NE Greece (two sites: one at 48 masl, OLY, and the other at 248 masl, STR, both with natural irrigation only) and Southern Germany (two sites: one site exposed to summer drought, SWA, and another one well-watered control site, RIE). The monitoring for survival rates 3 years after plantation showed low mortality in the German and the Italian sites and high mortality rates in the Greek sites triggered by extreme frost and drought events. In Greece and Italy, Q. pubescens was the best performing species, whereas in Germany, Q. pubescens and Q. robur performed best. The authors concluded that Mediterranean provenances (i.e., Greek, or Italian) might be used for future forestation objectives in Central Europe. Authors also remark that to ensure the success of seedling establishment in Quercus plantations in Northern Greece, water supply may be crucial.

To better understand the response of different oak species to changes in climate and the amount of water available, is certainly helpful to improve our predictions regarding their response to the ongoing climate change. However, I have some issues with this study that prevents me to provide a very positive response. Why the authors did not install a Mediterranean control site in Greece instead of two drought sites and a drought site in Italy paired with the installed control (SAN). As occurred in the Greek sites due to the frost and drought events, having a control site in other provenance not affected by these extreme climatic episodes don´t allow to compare between “treatments” properly.

  • Response: The manuscript was amended in order to qualify as a field performance report and provide an evaluation of the studied species’ establishment and growth in the common gardens during the first three years. Therefore, SAN and RIE are no longer considered as Control treatments. Regarding irrigation in the Greek sites, initially the authors intended to irrigate the seedlings but this was not possible due to technical problems such as blocked roads and the large distance in the finally available plantation site from available water. Irrigation at RIE and SAN occurred, whenever weather data indicated potential dry spells. Trees were then controlled and water was supplied, whenever first signs of drought occurred.

I have some minor concerns that should be addressed and clarified. I hope that my comments can help to clarify and improve some points.

Specific comments:

I miss some lines placing the novelty of the content and highlighting the significance and of the study in the abstract section.

  • Response, L15-18: The abstract was amended as suggested.

(L80-93) Along the introduction, when authors state the objectives of the study, in my opinion, they give too detailed information about sites and species that sounds quite redundant with 2.2 section.

  • Response, L92-100: In this paragraph, repetitions were removed as suggested.

(L33-161) The experimental design needs some clarification. The authors refer to a previous study for more detailed information, however, in my opinion it is complex enough to allow attaching a summary diagram of the set up.

Response, L141-177: A diagram of the setup of the replicate common gardens is attached as supplementary material (Figure S1).

Regarding to watering, maybe I misunderstood something, did you specify somewhere in the text the amount of water supply compared with the control? Did you supply by irrigation a specific amount simulating additional “normal” rainfall events respect to the control sites?

  • Response: Plantations were visually examined whenever there was a dry period and plants were irrigated whenever there were signs of drought events. The supplied water was not quantified.

(L138; L141) For the international audience specifying “near…” doesn´t add valuable information. If someone is interested in the specific location, she/he can head to coordinates.

  • Response, L141-162: This part was amended as suggested.

Results section: please review the verbal tense, sometimes is mixed present and past, authors must be consistent through the text.

  • Response: The results section was amended for greater verbal tense consistency as suggested.

(L256) Avoid subjective appreciations like “we can expect…” in the results section.

  • Response, L277: This phrase was removed as suggested.

(L330) Change the verbal tense for consistence “scores obtained in 2016 were considered”.

  • Response, L366 and Table 4: The term “score” was replaced with Relative Population Fitness (RPF).

I found the discussion better structured and explanatory than results section.

Being one of the main questions, concluding remarks on point 5 (predictive value of morphological and physiological measurements) are vague. 

  • Response, L533-535: The authors have now stated that the predictive value of root collar diameter is limited to only a few cases as it was discussed in the literature. Moreover, a decisive statement was included in the discussion section (L457-460).

Reviewer 3 Report

comments are attached

Author Response

The manuscript is well-written, and the authors have done a commendable job of characterizing the seedlings before and after planting for a number of variables.   There is, however, considerable bias from different sources that can affect early tree growth and make conclusions from this study problematic at this young age.  

Overall, the reviewer believes that a note, rather than a full paper, is more appropriate at this stage of growth. Although the author is North American and has no experience with these species, he has established plantations with over 15 oak species. In general, it takes three years for the root systems to fully develop and a balanced diameter/height growth to become established. He feels that that the plantation is too young to yield results to make specific conclusions. A note to inform the scientific community of preliminary results from this study would be useful and highly desirable. Some of the below biases will diminish over time as the seedlings recover from transplanting shock and begin to grow normally. The authors calculations of SPAD values at this early age will provide excellent baseline data for changes over time as the trees age. A detailed manuscript using age 5-7 results would be able to make more specific conclusions with a higher level of confidence.

  • Response: The manuscript was amended in order to qualify as a field performance report and provide an evaluation of the studied species’ establishment and growth in the common gardens during the first three years. Therefore, SAN and RIE are no longer considered as Control treatments, while the Discussion was amended accordingly.

Specific Comments:

The authors state that there were only 170 seedlings of Q. frainetto available for the Greek plantations.  Since the authors choose seedlings for planting according to their Plant Fitness scorn, the reviewer wonders if the 170 seedlings were chosen by their Fitness score, or was germination so poor that all seedlings were used, regardless of Fitness score?  If so, this could introduce bias in survival rates and perhaps growth over time. 

  • Response: Italian Q. frainetto in the Greek nurseries had low germination (now mentioned in L244-245), thus all seedlings were used regardless of their score. However, as visualized in Table 4, the survival rate of Q. frainetto at the Greek sites was also very low for the Greek provenance (stemming only 2 km from the plantation sites). Thus, the poor performance of Q. frainetto at OLY and STR was a common feature within the species.

The different seedling production methods could introduce bias in results at an early age.  This bias would disappear in surviving trees as the trees age over time.

  • Response: The authors agree with the comment. However, it was important to follow the local nursery protocols for the production of the specific species.

The soils are well described for each site, but there is no indication if they are typical oak sites for the species tested (reviewer is North American), how common are these sites in each country, e.g., 10% of oak forest sites in Germany, fit the description of the SWA site, and if the sites are cleared agricultural fields or post-timber harvest fields.  Seedling performance and survival in former agricultural field plantings is different than post-harvest sites for North American oak species.   The authors indicate that more information on the planting sites is in another publication, but this manuscript should stand alone and the most important details for the planting sites should be included.  Referring to the previous publication for more detailed information is acceptable, as long as the reader of this manuscript has enough information to decide if there is any overriding bias from site differences, e.g., one site was managed and another was not. 

  • Response, L141-177: Information about the former uses of each site and the experimental design was included in the manuscript.

There is no description of site preparation prior to planting nor post-planting management protocols. The manuscript mentions shading from trees at the SWA site and attributes better height growth due to the shading, which is true for young trees, but may be a detriment as the trees age.    Shading from weeds at the SAN site makes the reviewer question on whether post-planting management is consistent among sites.   If not, there will be a variable response in growth between and within sites, confounding comparisons of species growth rates and perhaps initial survival with site conditions.

  • Response, L159-161: Weed growth was intensive at SAN, SWA and RIE due to precipitation, thus the sites were cleared 2 to 3 times per year. It is now mentioned in the manuscript.

The authors’ description of the unfortunate frost event in Greece and resulting damage to the seedlings is thorough.  Surviving trees, over time, may recover from the damage to the root systems, depending on degree of damage and initial competition from other plant species.

Round 2

Reviewer 1 Report

The authors re-arranged the manuscript taking account of the issues I highlighted, and explaining in detail their reasons. The study is now presented in the form of a first report, with the perspective of future research to be done.

In view of the authors' efforts and of the comments by other two experienced reviewers, if the Editor finds it appropriate, I can retrace my steps and allow the manuscript's publication after some further adjustments, provided that the limits of the study are well stated in the text.

In particular, in the discussion, a part should be added where the flaws in the used methodology and their effects on the results are briefly explained, detailing the reasons that led to such flaws like the authors did in the response letter, to me and to the other reviewers.

Furhtermore, the title should state that this is a short-term/four-year study, e.g.: "Field performances of Mediterranean Oaks in replicate common gardens for future reforestation under climate change in central and southern Europe: results from a short-term study".

Author Response

The authors re-arranged the manuscript taking account of the issues I highlighted, and explaining in detail their reasons. The study is now presented in the form of a first report, with the perspective of future research to be done.

In view of the authors' efforts and of the comments by other two experienced reviewers, if the Editor finds it appropriate, I can retrace my steps and allow the manuscript's publication after some further adjustments, provided that the limits of the study are well stated in the text.

In particular, in the discussion, a part should be added where the flaws in the used methodology and their effects on the results are briefly explained, detailing the reasons that led to such flaws like the authors did in the response letter, to me and to the other reviewers.

  • Response, L356-379 and 388-390: A part showing the flaws of the study was included in the discussion, as suggested.

Furhtermore, the title should state that this is a short-term/four-year study, e.g.: "Field performances of Mediterranean Oaks in replicate common gardens for future reforestation under climate change in central and southern Europe: results from a short-term study".

  • Response: The title was amended as suggested.

Reviewer 3 Report

The authors have addressed my comments, however, I still feel that the manuscript should be greatly reduced in size and published as a note.   There are too many bias from differing nursery protocols to field treatments to make many conclusions in such a young planting.   I think it is important that the scientific community knows of the existence of the study, but an extensive analysis needs to come after the trees have grown for at least five years.  

Author Response

 The authors have addressed my comments, however, I still feel that the manuscript should be greatly reduced in size and published as a note. There are too many bias from differing nursery protocols to field treatments to make many conclusions in such a young planting. I think it is important that the scientific community knows of the existence of the study, but an extensive analysis needs to come after the trees have grown for at least five years. 

  • Response: We agree with the reviewer, that the results after three years of field performance should not be considered final answers to the underlying questions addressed. Therefore, it was intended to conduct measurements in 2020 (fifth growing year, fourth year in the common gardens) also. However, the 2020 season was completely lost due to COVID limitations on travel and field work abroad, especially with the lockdowns in Italy and Greece, and it is not clear yet, whether in the 2021 season we will be able to visit all sites. Concluding statements were therefore reduced. Moreover, answering to a comment by another reviewer, a paragraph addressing the flaws of the study was included in the discussion section (L356-379 and 388-390). The authors decided not to re-write the manuscript as a note (and submitting it to another journal with the option of publishing notes) since it has already been reviewed by two more reviewers for the journal Forests.